# Vascular Resection in Pancreatectomy—Is It Safe and Useful for Patients with Advanced Pancreatic Cancer?

**DOI:** 10.3390/cancers14051193

**Published:** 2022-02-25

**Authors:** Beata Jabłońska, Robert Król, Sławomir Mrowiec

**Affiliations:** 1Department of Digestive Tract Surgery, Medical University of Silesia, 40-752 Katowice, Poland; mrowasm@poczta.onet.pl; 2Department of General, Vascular and Transplant Surgery, Medical University of Silesia, 40-027 Katowice, Poland; robertk@hot.pl

**Keywords:** pancreatic cancer, borderline resectable, pancreatectomy, venous resection, arterial resection, vascular reconstruction, end-to-end anastomosis, interposition graft

## Abstract

**Simple Summary:**

Pancreatic ductal adenocarcinoma (PDAC) is a lethal disease with poor prognosis and increased incidence. Surgical resection R0 remains the most important treatment to prolong survival in PDAC patients. In borderline and locally advanced cancer, vascular resection and reconstruction during pancreatectomy enables achieving R0 resection. This study is a comprehensive review of the literature regarding the role of venous and arterial resection with vascular reconstruction in the treatment of pancreatic cancer.

**Abstract:**

Pancreatic ductal adenocarcinoma (PDAC) is a lethal disease with poor prognosis and increased incidence. Surgical resection R0 remains the most important treatment to prolong survival in PDAC patients. In borderline and locally advanced cancer, vascular resection and reconstruction during pancreatectomy enables achieving R0 resection. This study is a comprehensive review of the literature regarding the role of venous and arterial resection with vascular reconstruction in the treatment of pancreatic cancer. The literature review is focused on the use of venous and arterial resection with immediate vascular reconstruction in pancreaticoduodenectomy. Different types of venous and arterial resections are widely described. Different methods of vascular reconstructions, from primary vessel closure, through end-to-end vascular anastomosis, to interposition grafts with use autologous veins (internal jugular vein, saphenous vein, superficial femoral vein, external or internal iliac veins, inferior mesenteric vein, and left renal vein or gonadal vein), autologous substitute grafts constructed from various parts of parietal peritoneum including falciform ligament, cryopreserved and synthetic allografts. The most attention was given to the most common venous reconstructions, such as end-to-end anastomosis and interposition graft with the use of an autologous vein. Moreover, we presented mortality and morbidity rates as well as vascular patency and survival following pancreatectomy combined with vascular resection reported in cited articles.

## 1. Introduction

Pancreatic ductal adenocarcinoma (PDAC) is a lethal disease with a poor prognosis and increased incidence. According to literature, PDAC is the 14th most common cancer and the 7th highest cause of cancer mortality in the world [1]. Moreover, in two thirds of patients advanced or metastatic disease at the time of diagnosis is noted [2]. In addition, in 15–20% of patients with a potentially resectable PDAC undergoing pancreaticoduodenectomy (PD), the 5-year overall survival (OS) rate is 15–20% [2]. It should be noted that the common reason of the poor survival is non-radical (marginal-positive) resection. Additionally, it has been proven that margin resection status is a very important prognostic factor for OS and disease-free survival (DFS) in patients undergoing pancreatectomy for PDAC [2,3]. Therefore, assessing the resectability of the primary pancreatic tumor is very important for treatment planning. In 2006, Varadhachary et al. [2] first identified so-called borderline resectable (BR) pancreatic cancer as a distinct type of locally advanced pancreatic cancer [2]. It is known that abdominal and pelvic multidetector computed tomography (CT) is optimal for pancreatic radiological investigation to assess local tumor resectability [2]. According to consensus of International Study Group of Pancreatic Surgery (ISGPS) on BR pancreatic cancer, criteria for borderline resectability should be applied using a specialized pancreatic protocol CT performed in the previous 4 weeks including multidetector abdominal and pelvic CT [4] PDAC resectability depends on the tumor relationship to the adjacent large venous and arterial vessels such as portal vein (PV), superior mesenteric vein (SMV), superior mesenteric artery (SMA), hepatic artery (HA), celiac axis (CA). According to relationship to above-mentioned vessels, PDCA was divided into three resectability types: resectable, borderline resectable, and locally advanced (unresectable) [2,5]. Criteria for BR PDCA according to the National Comprehensive Cancer Network (NCCN) guidelines (version 2. 2014) [5] are presented in Box 1. In surgical practice, patients BR PDCA receive neoadjuvant chemotherapy, and those with stable or responding disease, by CT and serum tumor marker levels, then undergo pancreatectomy with potential necessity of concomitant vascular resection [2] Preoperative systemic therapy and/or local-regional chemoradiation is important to increase the probability R0 and to avoid R2 resection [6].

Box 1Criteria for borderline resectable pancreatic cancer according to the National Comprehensive Cancer Network (NCCN) guidelines.No distant metastases.Venous involvement of the superior mesenteric vein or
portal vein with distortion or narrowing of the vein or occlusion of the vein
with suitable vessel proximal and distal, allowing for safe resection and
replacement.Gastroduodenal artery encasement up to the hepatic artery
with either short segment encasement or direct abutment of the hepatic
artery, without extension to the celiac axis.Tumor abutment of the superior mesenteric artery not to
exceed greater than 180° of the circumference of the vessel wall.

The surgical assessment of the tumor resectability during pancreatectomy is also very important. The surgical technique and approach are very important to achieve negative margins and R0 resection. Sanjay et al. [7] described and summarized so-called “the artery-first approach” in pancreaticoduodenectomy (PD) which means that the artery is provided primary place in assessment of resectability, and arterial dissection is needed for the early assessment of arterial tumor involvement before performing an irreversible step in the surgical procedure. Different techniques for “artery-first approach” PD have been reported in the literature and they are summarized in the study by Sanjay et al. The authors described six approaches presenting various techniques for the early determination of arterial involvement, depending on the tumor location and size, and before the “point of no return” [7].

## 2. Methods of the Literature Search

The PubMed database (https://pubmed.ncbi.nlm.nih.gov/) was searched (the assessed date 12 January 2021–12 January 2022) for the following key words: “pancreatectomy” or “pancreaticoduodenectomy” and “vascular resection” or “venous resection” and “arterial resection”. The relevant original studies and meta-analyses were selected, reviewed, and discussed.

## 3. Venous Resections

### 3.1. Consensus of International Study Group of Pancreatic Surgery on BR Pancreatic Cancer

The International Study Group of Pancreatic Surgery strongly recommends surgical exploration and resection in the presence of possible for vascular reconstruction mesentericoportal axis involvement. According to these recommendations, venous resection is indicated if complete tumor excision (R0) is possible, although this may lead to a higher risk of intraoperative and postoperative complications. ISGPS also proposed classification of venous resections and distinguished four types of venous resections (Box 2) [4].

Box 2Types of venous resection according to International Study Group of Pancreatic Surgery.Type 1Partial venous excision with direct closure
(venorrhaphy) by suture closure.Type 2Partial venous excision using a patch.Type 3Segmental resection with primary veno-venous
anastomosis.Type 4Segmental resection with interposed venous
conduit and at least two anastomoses.

### 3.2. Literature Search for Types of Venous Resections in Pancreatectomy

The first case of PD with mesentericoportal vein (MPV) resection was reported by Moore et al. [8] in 1951 and then by Asada et al. [9] in 1963. In 1973, Fortner [10] proposed “regional pancreatectomy,” involving a systematic en bloc resection of the major peripancreatic vessels and wide soft-tissue clearance, in order to achieve R0 resection and to improve the long-term survival [11]. MPV resection in PDCA surgery has become a common procedure. Various vascular reconstructions using primary venous closure, end-to-end anastomosis, using autologous, allogenous or prosthesis graft were described in the worldwide literature. The overall patency rate of the above-mentioned reconstructions is between 70 and 90% [12,13,14,15,16,17,18]. Each technique has its own advantages and disadvantages. Therefore, the optimal venous reconstruction is still unclear.

### 3.3. Venous Reconstructions with the Use of Falciform Ligament and Other Parts of Parietal Peritoneum (“Safi Dokmak Vascular Graft”)

Recently, Shao et al. [12] retrospectively analyzed 146 patients undergoing MPV reconstruction between 2009 and 2018, including 13 FL (falciform ligament) venoplasty. The main aim of this study was to compare FL venoplasty with other types of venous reconstructions. Other reconstruction techniques included primary end-to-end anastomosis (primary, *n* = 30), lateral venorrhaphy (LV, *n* = 19), polytetrafluoroethylene conduit interposition (PTFE, *n* = 24), iliac artery (IA) allografts interposition (*n* = 47), and PV allografts interposition (*n* = 13). The not significantly shortest duration of postoperative hospitalization (*p* = 0.125), significantly shortest duration of operation (*p* = 0.023), and not significantly lowest blood loss (*p* = 0.109) were reported in the FL group. The patency rates of FL, primary anastomosis, LV, PTFE, IA, and PV reconstructions were 100%, 90%, 68%, 54%, 68%, and 85%, respectively. In comparison, FL had the highest patency rate (*p* = 0.008) and lowest antiplatelet/anticoagulation proportion (*p* = 0.000). Complications and long-term survival were comparable in all groups. The median survival time of patent group (24 months) was much longer, but not significantly (*p* = 0.148) compared with the thrombosed one (17 months 3) [12].

Three years prior, this author with co-authors [19], published report on six patients undergoing between June 2016 and May 2017 vascular resection during PD for malignant tumors. The FL graft, with a mean length of 26 mm (10–40), was immediately harvested and used as a lateral patch for reconstruction of the MPV. Anticoagulation therapy was not administered in any patients. Four complications were observed in two of six patients, including pleural effusion (*n* = 1), pancreatic fistula (*n* = 2), and bleeding (*n* = 1). The Clavien grade-III complications occurred in one (16.7%) patient but graft-related complications were not reported. Histological vascular invasion was present in all the patients (*n* = 6, 100%), and R0 resection was achieved in all cases (100%). A mean follow-up was performed at 12 (6–16) months, and the patency rate was 100% in this period. Authors concluded that an autologous FL graft was a safe lateral substitute for MPV reconstruction during PD and it improves the R0 resection rate of cancers [19].

In the other study, Zhiving et al. [20] described reconstructions of PV/SMV using a FL graft performed between May 2011 and July 2016, in 10 cases during pancreatectomy. These reconstructions involved a patch graft (*n* = 6) and a conduit graft (*n* = 4). In the early postoperative period, occlusion was noted in one case in a patient with a conduit graft, and stenosis in the other three cases with conduit graft was recognized 2 months following operation. Complete patency was noted in 3/6 patients with a patch graft, and stenosis in the other three cases 2 months following the surgery. Authors concluded that FL grafts might be considered for reconstruction of PV/SMV in the absence of appropriate vascular grafts [20].

In 2018, Malinka et al. [21] published a retrospective single-center analysis of 11 venous reconstructions with FL use in pancreatectomy performed between June 2017 and January 2018. There were nine wedge resections and two segmental resections in the analyzed group. Patch grafts were used for cases requiring only partial resection of the venous circumference, whereas interposition grafts were used for cases following segmental resections with tumor infiltration of >1/2 the venous circumference. Perioperative mortality rate was 0%. Unfractionated heparin was administered for 5 days, and it was followed by low molecular weight heparin (LMWH) to prevent deep vein thrombosis (DVT) during hospitalization. The patency was noted in 9/11 reconstructions during hospitalization and during the follow-up. This study revealed that the FL was a feasible autologous tissue for venous reconstruction with high postoperative patency [21].

Apart from FL, the so-called “Safi Dokmak Vascular Graft”, may be constructed from various parts of parietal peritoneum (PP), as the autologous substitute graft (ASG) used in venous resections during pancreatectomy [22,23]. In 2015, Dokmak et al. [22] published a study on 30 patients who underwent vascular resection during pancreatic (*n* = 18) or hepatic (*n* = 12) resection, mainly for malignant tumors (*n* = 29). Venous resection was an urgent procedure in four patients due to prolonged vascular occlusion. The PP, (a mean length of 22 mm (15–70)), was quickly harvested and used as a lateral (*n* = 28) or a tubular (*n* = 2) substitute for MPV reconstruction (*n* = 24), inferior vena cava (*n* = 3), or hepatic vein (*n* = 3). As perioperative anticoagulation, LMWH for 4 weeks was administered. The Clavien grade-III complications occurred in four (13%) patients. PP-related or hemorrhagic complications were not noted. Histological vascular invasion was present in 18 (62%) patients, and in all patients (100%) R0 resection was achieved. The mean follow-up was 14 (7–33) months. The patency rate was 97%. Authors showed that a PP can be safely used as a lateral patch for venous reconstruction during hepatobiliary surgery and to increase a rate of R0 resection [22].

In 2015, Dokmak [23] published a prospective study on a total of 52 patients who underwent hepatobiliary surgery with venous resection/reconstruction using PP. It was harvested rapidly through the same surgical incision in the same surgical field, and reconstruction was generally performed after the specimen was removed. The autologous substitute graft (ASG) was harvested from the PP of the diaphragm (*n* = 22), the hypochondrium (*n* = 19), the falciform ligament (*n* = 6), and the prerenal area (*n* = 5), and used as a lateral (*n* = 49) or tubular (*n* = 3) graft. Postoperative anticoagulation was used. Overall, 52 patients underwent pancreatectomy (*n* = 29), hepatectomy (*n* = 22), or both procedures (*n* = 1). The mean size of the ASG was 23 mm (range 10–80), and it was used for MPV reconstruction of the MPV (*n* = 42), the hepatic veins (*n* = 5), or the inferior vena cava (*n* = 5) for malignant neoplasm (98%). In six patients, urgent reconstruction was performed, due to prolonged vascular occlusion. One (2%) non-related mortality was noted as a result of septic complications after right hepatectomy. There were eight (15%) complications greater than grade III of the Clavien–Dindo classification. PP-related or hemorrhagic complications were not reported. The mean hospital stay was 16 (6–48) days. The mean follow-up was 11 (1–46) months. The patency rate was 96, 100% for the lateral grafts, and 33% for the tubular graft. The authors concluded that the Safi Dokmak vascular graft using the PP for lateral reconstruction of the MPV can be harvested rapidly with no limitation in size. Moreover, it is inexpensive and safe, therapeutic anticoagulation following surgery is not necessary and the theoretical risk of infectious complications is very low [23].

#### Conclusions

PP, including FL, can be a useful, safe, and inexpensive alternative method for the venous reconstruction in pancreatectomy. It can be rapidly harvested, and it can be used in the elective and emergency procedures. The short-term and long-term results, including overall patency, are very good.

### 3.4. Review of Different Venous Reconstructions Types including End-to-End Anastomosis and Interposition Graft with the Use of Autologous Vein

In 2010, Lee et al. [24] published a study on 34 patients undergoing PV/SMV reconstruction during PD using veins of the lower extremity. The great saphenous vein (GSV) was preferred for patching and the femoral vein (FV) for replacement. The FVs were harvested from the mid to proximal thigh up to the profunda femoris vein and GSVs distally from the saphenofemoral junction. PV/SMV replacement using FV was performed in 15 patients. The complications associated with the vein harvesting were as follows: in 7/15 patients, small postoperative lower extremity edema was noted, and in 5 patients- wound complications secondary to the FV harvest place were observed. Among 15 patients, with PV/SMV patching with the use of GSV, there were no postoperative wound complications, and minimal postoperative lower extremity edema was reported in one patient. Among four patients receiving PV/SMV patching with the use of FV, there were no postoperative wound complications, and minimal postoperative lower extremity edema was reported in one patient. Long-term survival was comparable in patients undergoing PD without PV/SMV reconstruction and patients with venous reconstruction. Authors concluded that the PV/SMV reconstruction using leg veins is associated with good patency with minimal postoperative lower extremity complications and not increased late mortality. In the authors’ opinion, the lower extremities should be routinely included in the operative field of patients undergoing PD [24].

In 2012, Turley et al. [25] published a study involving 204 patients undergoing PD for PDAC from 1997 to 2008. Patients undergoing PD with vascular reconstruction (VR) (*n* = 42) were compared with patients undergoing standard PD (*n* = 162). VRs were performed by a vascular surgeon and involved primary repair (*n* = 8), vein patch (*n* = 25), or interposition grafting (*n* = 9) with femoral, external iliac, inferior mesenteric, or saphenous veins. Routine thromboprophylaxis and acetyl salicylic acid (ASA) were used in the perioperative period. In patients undergoing PD with VR, larger tumors (3.0 cm vs. 2.5 cm, *p* < 0.01) were reported. Rates of tumor-free margins (73% vs. 72%, *p* = 0.84) or lymph nodes metastases (50% vs. 38%, *p* = 0.14) were comparable in both groups. In the VR group, higher median blood loss (875 mL vs. 550 mL, *p* < 0.01) was noted. The mortality and morbidity rates, duration of hospitalization, and readmission rates were similar. A median follow-up was 29 months. Graft patency was 91.7%. OS rates were similar in both groups. In multivariate analysis, poor prognostic factors for survival were as follows: higher histological grade (*p* = 0.01), lymph nodes metastases (*p* = 0.01), and larger tumor size (*p* = 0.01). VR (*p* = 0.28) was not a poor prognostic factor for survival. The patency rate was 97%. Based on the above-mentioned results, the authors concluded that the need for VR was not a contraindication to potentially curative resection in PDAC patients and VR should be performed with the assistance of a vascular surgeon [25].

In 2014, Krepline et al. [26] published the results of 43 pancreatectomies with venous reconstructions including the following: tangential resection with primary repair (*n* = 7) (16%) or GSV patch (*n* = 9) (21%); segmental resection with splenic vein division and either primary anastomosis (*n* = 10) (23%) or internal jugular vein interposition (*n* = 8) (19%); or segmental resection with splenic vein preservation and either primary anastomosis (*n* = 3) (7%) or interposition grafting (*n* = 6) (14%). ASA was administered in all patients following surgery while LMWH was not routinely administered. The median follow-up was 13 months. In this period, in four (9%) patients, vascular occlusion was noted. The median duration to detection of thrombosis in the four patients was 72 (16–238) days [26].

In 2014, Hirono et al. [27] reported 128 pancreatic resections (5 total pancreatectomies (TP), 99 PD, and 24 distal pancreatectomies (DP)) with PV/SMV resection, including 14 with the grafts use. The criteria for the venous resection type were as follows: if tumor involvement >1/4 of the PV/SMV wall was suspected, a tangential PV/SMV resection was performed; and for larger venous involvement-circumferential resection was performed. In 14 patients undergoing PV/SMV reconstruction with grafts, the grafts were harvested from the external iliac vein (EIV) in (*n* = 10) and internal jugular vein (IJV) (*n* = 4). An intraoperative or postoperative acute thrombus or stenosis of reconstructed PV/SMV, after direct end-to-end anastomosis, was reported in five (3.9%) patients. However, PV/SMV patency was excellent after reconstruction using grafts. Complications rates were comparable in both groups. In three (30%) patients with EIV grafts, postoperative leg edema was noted, and one patient died due to leg compartment syndrome. There were no complications in patients with IJV grafts. This study showed that IJV use for venous reconstruction was superior compared with EIV, because it was not associated with any complications [27].

In 2015, Glebova et al. [28], based on a prospective study involving 173 patients undergoing pancreatectomy with portal vein reconstruction (PVR) from 1970 to 2014, stated that the long duration of operation and use of prosthetic grafts for venous reconstruction were risk factors for postoperative PV thrombosis. In this study, primary lateral venorrhaphy was performed if the degree of luminal narrowing was <30%. If the PV lumen narrowing was >30%, primary end-to-end anastomosis was performed (in possibility of PV mobilization), or a repair with a patch or interposition graft. Primary anastomosis was performed usually if the length was <2 cm. In a larger length, an interposition graft was used. For vein interposition, internal jugular (4), left renal (4), splenic (2), and great saphenous (1) veins were used. For synthetic grafts, 8-mm Dacron (1) and PTFE (four ringed and one non-ringed) were used. In the authors’ opinion, primary repair, patch, or vein interposition should be preferentially used for PVR in the setting of pancreatic resection [28].

Different venous reconstruction types were also analyzed and compared in the study by Dua et al. [29] involving 90 patients undergoing PV/SMV resections during a pancreatectomy (PD, TP, DP) from 2005 to 2014. The following reconstruction techniques were performed: (1) longitudinal venorrhaphy (LV, *n* = 17); (2) transverse venorrhaphy (TV, *n* = 9); (3) primary end-to-end (*n* = 28); (4) patch venoplasty (PV, *n* = 17); and (5) interposition graft (IG, *n* = 19). The study showed that the thrombosis rate was significantly associated with venous reconstruction technique. The patency rate of primary end-to-end anastomosis or TV was 100%. LV, PV, and IG were all associated with significant rates of thrombosis (*p* = 0.001). The patency of primary end-to-end anastomosis and TV is higher compared with the other alternatives; therefore, these techniques should be preferred for short (<3 cm) reconstructions [29].

In 2019, Terasaki et al. [30] reported 199 PD including 122 PD with PVR between 2001 and 2017. The end-to-end anastomosis was performed in 97 (79.5%) patients, and an interposition graft using the right external iliac vein (REIV) was performed in 25 (20.5%) patients. The 2-year and 5-year survival rates of the no-PVR group (54.2% and 30.8%, respectively) were significantly longer compared with both the end-to-end anastomosis group (24.5% and 13.7%) and the interposition graft group (32% and 10.0%) (*p* < 0.001). Survival rates in patients following the end-to-end anastomosis and the interposition graft were comparable (*p* = 0.963). Therefore, authors concluded that an interposition graft using the REIV for PVR following PD was safe and effective [30].

In 2020, Pantoya et al. [15] published a study on 18 patients comparing two methods of venous reconstruction with graft use following PD with venous resection in 2011–2019. The two veins for an interposition graft were used and compared. GSV was used in 13 patients, and IJV was used in 5 patients. The authors concluded that both reconstruction methods were comparable [15].

Generally, a long-term patency of venous reconstruction (using primary closure/anastomosis or interposition graft) is very important in patients following a pancreatectomy combined with venous resection. It can be said that such as from an oncological point of view, R0 resection is important, venous patency is essential from a vascular point of view. According to the literature, the overall rates of occlusion of PV are 0–17% and it depends on the extent of resection, and the method and timing of reconstruction [31]. In 2020, Chan et al. [31], showed that a 1-year primary venous patency of primary repair was superior to end-to-end anastomosis and interposition graft. In this study, primary repair was performed in 47, end-to-end anastomosis in 19 patients, and interposition graft in 10 patients. The authors added that an interposition graft should be still performed in cases when primary vein closure or anastomosis are not possible, such as an extensive length of venous resection [31].

In 2020, Labori et al. [32] published a systematic review on the optimal graft type for SMV and PV in pancreatic surgery, including papers published between January 2000 and March 2020 (34 studies with 603 patients). Four graft types were identified (autologous vein, autologous parietal peritoneum/falciform ligament, allogeneic cadaveric vein/artery, synthetic grafts). The early and overall graft thrombosis incidence was 7.5% and 22.2% for synthetic graft, 5.6% and 11.7% for autologous vein graft, 6.7% and 8.9% for autologous parietal peritoneum/falciform ligament, and 2.5% and 6.2% for allograft. Complications located in the donor site were noted for harvesting of the femoral, saphenous, and external iliac veins. There was no graft infection in synthetic grafts. Authors concluded that autologous, allogenic or synthetic grafts for SMV-PV reconstruction were safe and feasible in selected patient groups. A higher rate of graft thrombosis was noted in synthetic grafts. This systematic review confirmed that there was no consensus on the definition and reporting of long-term graft patency following pancreatectomy with SMV-PV resection and reconstruction. Authors noted that there was no consensus on anticoagulation management in the postoperative period in patients following pancreatectomy with venous resection. In this meta-analysis, studies regarding the use of a cadaveric vein or arterial allografts and synthetic PTFE grafts in patients following venous resection during pancreatectomy were widely analyzed and discussed. It should be noted that above-mentioned graft types are less frequently used in planned procedures. It is known that autologous vein harvesting for venous interposition graft is associated with an additional surgical procedure performed at another site of the body, and prolonged duration of operation. Cadaveric allografts and PTFE grafts can save operative time that is very important especially in unplanned venous resections and in patients in a poor general condition in whom operating time is essential. In planned operations, some surgeons prefer to avoid synthetic grafts due to a higher risk of thrombosis and infection compared with native grafts. Labori’s meta-analysis confirmed that graft thrombosis is more frequent after synthetic graft reconstruction (which is also obvious in vascular surgery concerning peripheral arteries disease (PAD)). Although graft thrombosis can occur in all graft types (early incidence is 2.5–7.5%, and overall incidence is 6.2–22.2%). It should be added that the association of PTFE grafts with increased mortality was not shown in this meta-analysis [32]. In order to reduce a rate of infectious complications associated with PTFE use, various methods, such as administration of perioperative antibiotics, graft immersion in antibiotic solution, avoidance of enteric and biliary spillage, placement of an omental wrap, and use of antibiotic irrigant, were reported in the literature [13,32]. This study showed that generally most surgeons preferred a segmental resection with primary anastomosis or a partial venous excision with direct closure due to the higher risk of complications following the use of an interposition graft [32]. We would like to add that the possibility of segmental resection with primary vein closure or primary end-to-end anastomosis depends on the length and circumference of tumor infiltration which determines the venous resection type. Therefore, we can conclude that a risk of complications is proportional to PDCA venous extension. In the other words, the greater the infiltration of the vein, the greater the extent of the venous resection and the more complicated vascular reconstruction, and thus the greater the risk of complications.

Regarding the above-mentioned resection length, in 2013, Pan et al. [33] compared postoperative outcomes in PDAC patients undergoing PD with or without superior mesenteric vein/portal vein resection (SMV/PVR) in relation to vein resection length. This study involved 118 patients who underwent surgery between 2005 and 2010. SMV/PVR was performed in 58 patients. Among them, in 28 patients, the length of SMV/PVR was ≤3 cm, whereas in the other 30 patients, the length of SMV/PVR was >3 cm. Short-term results such as operative time (435 min vs. 477 min, *p* = 0.063), blood loss (300 mL vs. 383 mL, *p* = 0.071), and transfusion volume (85.7 mL vs. 166.7 mL, P = 0.084) were comparable in two groups. There was a significant difference between the two groups (SMV/PVR ≤ 3 cm and SMV/PVR > 3 cm) in terms of the mean survival time (18 months vs. 11 months) and the overall 1- and 3-year survival rates (67.9% and 14.3% vs. 41.3% and 5.7%, *p* < 0.02). Concluding, this study showed significantly better survival in patients with shorter resections, but short-term postoperative results were comparable regardless of venous resection length. In the authors’ opinion, worse survival in patients following longer venous resections is associated with a higher number of adverse oncological factors (such as the ratio of lymph nodes invasion). We should add that in this study, authors achieved tension-free end-to-end anastomosis by dissociating the root of the SMV when the length of PVR was about 5 cm [33]. Some other institutions have suggested that a distance of up to 8 cm can achieve primary anastomosis by this procedure [34,35].

Concerning a safe length of venous resection with primary anastomosis, in 2015, Fujii et al. [36] analyzed retrospectively 810 patients undergoing pancreatectomy (involving 197 SMV/PV resections) from January 2000 to April 2014. The development of severe anastomotic stenosis (≥70% occlusion) was analyzed. Among patients with no recurrence in the 1-year follow-up period, stenosis was observed in 18 patients. In a univariate analysis, operation time ≥ 520 min and length of SMV/PV resection ≥ 31 mm were associated with the development of severe anastomotic stenosis. In a multivariate analysis, the length of SMV/PV resection ≥ 31 mm was an independent predictor of severe anastomotic stenosis (Hazard Ratio, HR = 5.96; *p* = 0.003). This study showed that direct SMV/PV end-to-end anastomosis is safe and effective for R0 resection. If tension-free anastomosis cannot be guaranteed, generally following SMV/PV resection of length ≥ 31 mm, an autologous graft may be needed in order to prevent anastomotic stenosis [36].

A study by Kim et al. [37] showed that planned PV resections for PDAC are associated with higher rates of postoperative major and vascular complications and higher R0 resection rates compared with unplanned resections. Authors analyzed PDs performed in the period of 11 years. Patients undergoing PV resection (PV-PD) were selected and divided into two groups: undergoing planned or unplanned resection. Of 249 patients, 66 (26.5%) had PV-PD, including 27 (41%) planned and 39 (59%) unplanned PV resections. Planned PV resections were performed in significantly younger patients (60 ± 9 vs. 65 ± 10 years; *p* = 0.031), and associated with longer operating times (602 ± 131 vs. 458 ± 83 min; *p* < 0.001) and more major complications (26% vs. 5%; *p* = 0.026). Planned PV resections were associated with a lower rate of positive margins (4 vs. 44%; *p* < 0.001) despite a larger tumor size (3.9 ± 1.4 vs. 2.9 ± 1.0 cm; *p* = 0.002). Survival rates were comparable in two groups (*p* = 0.998) [37].

#### Conclusions

The type of venous reconstruction in pancreatectomy with concomitant vascular resection depends on the tumor infiltration involving the MPV. According to most authors, the highest patency rate is noted in primary venorrhaphy followed by segmental resection, interposition autologous vein graft and synthetic PTFE grafts. The possibility of partial venous excision with primary vein closure or segmental resection with primary end-to-end anastomosis depends on the length and circumference of tumor infiltration which determines venous resection type. In infiltration with vascular lumen narrowing after a resection <30%, primary vein closure is possible. When it is not possible, segmental resection with end-to-end anastomosis is needed. According to most authors, primary end-to-end anastomosis is possible in resection involving <2–3 cm of the venous length. An interposition graft should be performed in cases when primary vein closure or anastomosis are not possible. The autologous patient’s vein is the graft of choice. We prefer GSV for such reconstruction, because it is associated with a lower risk of potential complications related to graft harvesting. When using the autologous vein is not possible, the synthetic graft using PTFE is recommended. PP is the interesting alternative for venous reconstructions, because its preparation does not need to harvest a vascular graft, which is associated with the shorter duration of operation. Currently, there is no consensus and clear guidelines regarding perioperative anticoagulation in patients undergoing pancreatectomy with concomitant venous resections, it should be precise.

### 3.5. Sinistral (Left-Sided) Portal Hypertension Following Mesentericoportal Resection without Reconstruction of Splenic Vein

Ligation of the splenic vein during PD can cause sinistral (left-sided) portal hypertension (LPH) leading to variceal bleeding and thrombocytopenia by hypersplenism [38,39,40,41]. The incidence of clinically relevant sinistral portal hypertension is uncommon. Tanaka et al. [38] showed that the preservation of so-called critical veins, such as superior right colic vein (SRCV) arcade, middle colic vein (MCV), and left gastric vein (LGV) is useful to decrease the LPH risk. In the authors’ opinion, the indication for splenic vein reconstruction should be tailored according to individual risk factors [38]. Ono et al. [39] suggested to preserve the right colic marginal vein to avoid the development of varices. In the authors’ opinion, reconstruction of the splenic vein should be considered if the right colic marginal vein is divided during operation [39]. Addeo et al. [40] proposed the left splenorenal shunt in order to decrease clinical signs of left portal hypertension following ligation of the splenic vein and inferior mesenteric vein confluence preservation [40]. According to Gyoten et al. [41], simultaneous ligation of a splenic artery with a vein is an another way to avoid results of sinistral portal hypertension following PD [41].

### 3.6. Short-Term Results and Survival Following Pancreatectomy with Venous Resection

Regarding survival following pancreatectomy with venous resection compared with a pancreatectomy without it, numerous studies were published. In 2014, Cheung et al. [42] published a comparative study carried out on 78 PDCA patients undergoing PD between 2001 and 2012. Among them, 46 patients received standard PD (group 1) and 32 patients received PD with simultaneous resection of the portal vein or the superior mesenteric vein or artery (group 2) followed by reconstruction. In this study, the morbidity and mortality rate were comparable in both groups. The long-term results were also similar. The 1-year, 3-year, and 5-year OS rates in group 1 were 71.1%, 23.6%, and 13.5%, and in group 2 were 70.6%, 33.3% and 22.2% respectively (*p* = 0.815) [42].

Selvaggi et al. [43] compared 40 patients undergoing pancreatectomy and venous resection (VR) (group A), and 20 patients (group B) undergoing bilio-enteric and/or gastro-entero bypass. The median survival was 25 months after PV resection and 17.8 months after palliative surgery. Their results showed that VR can determine a better oncological result after the first 12 months from surgical resection rather than surgical palliation alone [43].

In 2014, Yu et al. [11] published a meta-analysis involving studies comparing PD with venous resection (VR) vs. PD without venous resection (WVR) groups. This meta-analysis involved 22 retrospective studies including 2890 patients. In this meta-analysis, perioperative morbidity, mortality, and 1-year and 3-year survival were comparable in the two groups. There was a statistical difference in median tumor size (*p* < 0.001), R0 resection rate (*p* < 0.001), lymph node metastasis (*p* = 0.03), pancreatic fistula (*p* = 0.01), and 5-year survival (*p* = 0.03) between the two groups. Additionally, in patients following PD with venous resection R0, a significantly better 2-year (*p* < 0.001) and 5-year (*p* = 0.00002) survival compared with patients receiving R1 resection was noted. Authors concluded that PD with venous resection can achieve equal perioperative morbidity and mortality as standard resection. However, R0 resection should be performed as far as possible for optimal survival [11].

In a multi-center study by Murakami et al. [44], patients undergoing PD with or without venous resection were compared. Among 937 patients undergoing PD, 435 (46.4%) had PV/SMV resection, whereas the remaining 502 (53.6%) had standard PD. Perioperative mortality and morbidity were comparable in both groups. In a multivariable analysis, PV/SMV resection was not an independent prognostic factor for OS (*p* = 0.268). Among the 435 patients undergoing PV/SMV resection, borderline resectable tumors with arterial abutment (*p* = 0.021) and absence of adjuvant chemotherapy (*p* < 0.001) were independent predictors of poor survival in multivariable analysis. A median survival time was 43.7 and 29.7 months in patients with resectable or borderline resectable tumors with PV/SMV involvement, respectively. Median survival time in patients with borderline resectable tumors with arterial abutment was 18.6 months despite adjuvant chemotherapy [44].

Jeong et al. [45] in a study conducted on 276 patients undergoing PD between 1995 and 2009, showed that short-term and long-term results in patients with PV-SMV resection and without PV-SMV resection are comparable. In this study, 46 patients (16.7%) underwent PV-SMV resection during PD. Postoperative severe morbidity (grade 3 or 4) was similar for patients with and without PV-SMV resection (8.7% with, vs. 7.0% without *p* = 0.754). The mortality rate was also similar: 2.2% with PV-SMV resection and 0.9% without PV-SMV resection (*p* = 0.423). Survival of PV-SMV resection and no resection group were also comparable in the two groups (median survival, 16 vs. 12 months; *p* = 0.086). Additionally, there was not a significant difference in OS between patients with and without pathologic PV-SMV invasion (median survival, 13 vs. 16 months; *p* = 0.663) [45].

Wang et al. [46] reported a retrospective analysis of 208 patients undergoing PD (PD group) involving PD with PV/SMV resection and reconstruction (PDVR group) for PDAC from 2009 to 2013. In the PDVR group (42 patients), the 1-, 2-, and 3-year survival rates were 70%, 41%, and 16%, respectively, and the median survival time was 20.0 months. In the PD group (166 patients), the 1-, 2-, and 3-year survival rates were 80%, 52%, and 12%, respectively, with the median survival time of 26.0 months. Survival time and R0 resection ratio were comparable in both groups [46].

In 2015, Delpero et al. [47] published a multicenter study involving 1399 patients undergoing PD or TP for PDAC, from 2004 to 2009, with or without VR (997 standard resections [SR]) or with VR (402 patients; 29%). Postoperative morbidity and mortality were comparable in the two groups (5 vs. 3 % in SR patients; *p* = 0.16). The median and 3-year survival rates in VR patients vs. SR patients were 21 months and 31 % vs. 29 months and 44%, respectively (*p* = 0.0002). In a multivariate analysis, VR was a significant poor prognostic factor for long-term survival (HR = 1.75; *p* = 0.0005).

In 2019, Serenari et al. [48] compared short-term and long-term results following four types of pancreatic resections involving two pancreatic resection types: PD and TP and two venous resection types: tangential (TVR) and segmental venous (SVR) resections: (1) PD + TVR, (2) PD + SVR, (3) TP + TVR, (4) TP + SVR. Simultaneous pancreatic and venous resections for PDAC were performed in 99 patients. Among them, in 25 PD + TVR (25.3%), 12 − PD + SVR (12.1%), 23 − TP + TVR (23.2%), and 39 − TP + SVR (39.4%) were performed. Overall, the major morbidity (Clavien–Dindo grade ≥ IIIA) was 26.3%. The 30- and 90-day mortality were 3% and 11.1%, respectively. Short-term results were similar in all groups. The median OS of patients undergoing PD + TVR was significantly higher than those to TP+SVR (29.5 vs. 7.9 months, *p* = 0.001). The independent prognostic factors for OS in a multivariate analysis were TP (HR = 2.11; *p* = 0.002) and SVR (HR = 2.01; *p* = 0.003) [48].

In 2016, Giovinazzo et al. [49] published a meta-analysis including 27 studies involving 9005 patients (1587 in pancreatectomy with PV-SMV resection group). In patients undergoing PV-SMV resection, an increased risk of postoperative mortality (*p* = 0.2) and of R1/R2 resection (*p* < 0.001) compared with patients undergoing standard pancreatectomy was noted. The 1-, 3-, and 5-year survival were worse in the PV-SMV resection group: HR = 1.23 (*p* = 0.005), HR = 1.48 (*p* = 0.004), and HR = 3.18 (*p* < 0.001), respectively. The median OS was 14.3 months for patients undergoing pancreatic resection with PV-SMV resection and 19.5 months for patients without vein resection (*p* = 0.063). This meta-analysis showed increased postoperative mortality, higher rates of non-radical surgery, and worse survival after pancreatic resection with PV-SMV resection. In the authors’ opinion, these results were associated with more advanced disease in the venous resection group [49].

In 2019, Peng et al. [50] published a meta-analysis on patients undergoing PD with venous resection (PDVR). This meta-analysis included 30 studies comprising 12,031 patients (2186 who underwent PDVR and 9845 who underwent PD). PD and PDVR groups were compared. In the PDVR group, a lower R0 resection rate and higher rates of complications such as biliary fistula, reoperation rate, delayed gastric emptying, cardiopulmonary abnormalities, hemorrhage, in-hospital mortality, and 30-day mortality were noted. In addition, the blood loss, duration of operation and hospitalization were higher in the PDVR group. This meta-analysis showed that PDVR was associated with a higher risk of morbidity and mortality, longer duration of hospitalization, and a lower R0 resection rate. Therefore, in the authors’ opinion, PDVR for pancreatic cancer should be carefully selected by the surgeon [50].

In 2020, Fancellu et al. [51] published a meta-analysis involving 23 cohort studies including 6037 patients, of which 28.6% underwent PD + VR and 71.4% underwent standard PD. In this study, 30-day mortality was significantly higher in the PD + VR group (OR 1.93, *p* = 0.002), although morbidity rates were comparable in the two groups (OR 1.07, *p* = 0.65) In patients following PD + VR, lower 1-year OS (odds radio (OR) 0.79, *p* = 0.003), 3-year OS (OR 0.72, *p* = 0.0006), and 5-year OS (OR 0.57, *p* = 0.003) were noted [51].

A recent meta-analysis by Filho et al. [52] confirmed that PDVR was associated with a higher risk for postoperative morbidity and mortality, longer duration of hospitalization, and higher blood loss as well as a worse OS compared with standard PD.

#### Conclusions

There are different short- and long-term results of PDVR reported in the literature. According to some single-center retrospective studies, short-term and long-term results are comparable in standard PD and PDVR. According to other authors, OS is worse in PDVR. Three meta-analyses showed worse short-term and long-term results in PDVR which can be associated with a higher PDCA advancement in patients undergoing PD with venous resection.

## 4. Arterial Resections

### 4.1. Consensus of International Study Group of Pancreatic Surgery on BR Pancreatic Cancer

There is no good evidence that arterial resections during PD are associated with benefits for patients. Such resections may be associated with increased morbidity and mortality and should not be recommended on a routine basis. Patients categorized as BR on the basis of features of arterial involvement in CT, should undergo exploratory laparotomy in order to obtain further verification of any arterial infiltration. Currently, in case of verification of arterial involvement, resection is not indicated [4].

So, currently, according to ISGPS consensus, there is no recommendation for arterial resection during PD. While concurrent venous resections are routinely performed and currently are standard procedures, arterial resections are still controversial [53].

### 4.2. Literature Search for Arterial Resections in Pancreatectomy

In 2010, Ouaissi et al. [54] analyzed the outcomes of 149 patients undergoing pancreatectomy (PD (136, 91.28%), TP (13, 8.72%)) without vascular resection (group A: 82 patients), with isolated venous resection (group B: 67 patients), or with arterial (SMA, CHA, right hepatic artery (RHA)) and/or venous resection (group C: eight patients). All eight (100%) patients in group C underwent PD. In order to anticipate vascular resection and venous graft harvesting, if it was needed, the left cervical and bilateral groin areas were routinely included into the operative field draping. The authors preferred end-to-end vascular anastomosis in case of limited invasion while an autologous venous graft was used if the venous resection exceeded 5 cm in length. It this study, splenic vein (SV) was routinely re-implanted into the superior mesenteric vein (SMV) or the portal vein (PV) in order to avoid sinistral hypertension. Arterial resection concerned SMA in one patient, the CHA in two patients, and a right anomalous hepatic artery in five patients. Arterial reconstruction was as follows: six (75%) end-to-end anastomoses and two (25%) venous interposition grafts using GSV. Generally, venous graft was used in 23 patients, including left internal jugular vein in 14 patients and saphenous vein in 8 patients. A significantly higher duration of operation and blood loss was noted in groups B and C compared with group A. It is obvious that the longest operative time (570 [420–600] min) was recorded in patients undergoing arterial resection (*p* = 0.025). Postoperative morbidity and mortality rates, including bleeding, were comparable in all groups (*p* = 0.44). Postoperative hospital stay was also comparable in all groups (*p* = 0.642). R1 resection was significantly more frequent in groups B (42%) and C (50%) compared with group A (13%; *p* = 0.0002). It was associated with more advanced tumors in these groups. Ten-year overall and disease-free survivals were significantly better in group A (19 and 20%) compared with group B (2.8 and 0%) and group C (0% and 0%). Concluding, this study revealed that vascular resection combined with PD for PDAC increases local resectability without increasing mortality and morbidity rates but does not improve patients’ survival [53].

In 2011, Bockhorn et al. [55] analyzed results of arterial en bloc resection (AEBR) in PDAC patients. Patients were divided into three groups: 29 patients undergoing pancreatectomy and AEBR (group 1), 449 undergoing pancreatic resection without arterial resection or reconstruction (group 2), and 40 with unresectable tumors undergoing bypass (group 3). There were the following operation types in group 1: PD, 15 (52%); pylorus-preserving pancreaticoduodenectomy (PPPD), 1 (3%); distal pancreatectomy with splenectomy, 5 (17%); subtotal pancreatectomy, 4 (14%); total pancreaticoduodenectomy with splenectomy, 4 (14%); and additional multivisceral resection, 19 (66%). Procedures in the AEBR group were as follows: 16 pancreaticoduodenectomies (including 1 pylorus preserving PD), 4 subtotal pancreatectomies, 5 distal pancreatectomies with splenectomy, and 4 total pancreatoduodenectomies with splenectomy. Arterial resections included CA, CHA, and SMA. Eighteen patients underwent reconstruction of the hepatic artery. Vascular reconstruction was performed in an end-to-end anastomosis (10) and in remaining eight patients: using an interposition graft of cryopreserved blood type-matched vessel (6), saphenous vein (1), or inferior mesenteric vein (1). Resection of the celiac trunk was performed in eight patients. An interposition cryopreserved graft from either the aorta (3) or SMA (4) was used to reconstruct the proper hepatic artery in seven patients; in the remaining instance a saphenous vein graft was interposed from an aberrant left hepatic artery to the proper hepatic artery. Cryopreserved vessels were used for reconstruction in three patients in whom resection of the SMA was required. All patients undergoing arterial resection received systemic anticoagulation starting approximately 2 h after operation to achieve a partial thromboplastin time of 60–70 s. In remaining patients, conventional prophylactic heparinization with LMWH was administered. Perioperative morbidity and mortality rates were higher in group 1 compared with group 2 (*p* = 0.031 and *p* = 0.037 respectively). Additional portal vein resection was an independent predictor of morbidity (*p* < 0.001). There were 6 minor and 15 major complications among 11 patients in group 1, with a relaparotomy rate of 21% (6/29). There were four postoperative deaths from surgical complications in group 1 (mortality rate 14%). This was significantly higher than rates of 4.0% (18/449) in group 2 (*p* = 0.037) and 5% (2/40) in group 3 (*p* = 0.230). Duration of operation was significantly longer in group 1 (350 (220–480) min) compared with groups 2 and 3 (*p* = 0.001). Hospital stay was the longest in group 1 (24 (18–115)) (*p* = 0.049). The median OS was comparable in two groups (14.0 vs. 15.8 months; *p* = 0.152), and lower for group 3 (7.5 months; *p* = 0.028 vs. group 1). This study confirmed significantly higher morbidity and mortality following arterial resection. Authors concluded that in selected patients, AEBR can result in OS comparable to standard resection and significantly better compared with OS after palliative bypass [54].

In 2011, Bachelier et al. [56] published a study on arterial resections in pancreatectomy. This study included 52 patients divided into two groups: the study group AR + (*n* = 26) and the control group AR − (*n* = 26). There were 21 PD and 5 DP in patients with arterial resection. Patients underwent pancreatectomy between 1990 and 2008. In this study, arterial reconstruction was performed in 15 patients: using a direct tension-free arterial anastomosis (7), interposition graft (5), complex arterial reconstructions including direct tension-free arterial anastomosis and the use of an interposition graft (2). Among five patients requiring an interposition graft, in two cases, authors used the splenic artery which was returned to perform end-to-end anastomosis with the hepatic artery. A prosthetic interposition graft was performed in only one patient. In 11 patients, the arterial reconstruction was not performed due to the presence of an efficient collateral arterial circulation for the liver. Short-term results were comparable in two groups. Duration of operation was similar in both groups: 518.99 (345–705) vs. 440.110 (165–600) min (*p* = 0.112). Perioperative morbidity (*p* = 0.264) and mortality (*p* = 1.0) were similar. Regarding the long-term results, the 1- and 3-year survival rates were comparable in two groups (65.9% and 22.1%, median 17 months, for the group AR+, vs. 50.0% and 17.6%, median 12 months, for the group AR−; *p* = 0.581). The multivariate analysis showed that the arterial wall invasion at the site of AR, the total number of resected lymph nodes of ≤15, and perineural invasion were independent prognostic factors for survival [55].

In 2020, the same first author Bachelier et al. [57] reported a single institution experience with 118 patients undergoing pancreatectomy with arterial resection for PDAC between 1990 and 2018. In our opinion, this article is the most important concerning arterial resections in pancreatectomy, because recently it is the largest single-center study regarding pancreatectomy with arterial resection. Therefore, it is widely discussed in our review. The following pancreatic resection types were analyzed in this study: 51 pancreaticoduodenectomies, 18 total pancreatectomies, and 49 distal pancreatectomies with splenectomy. Resected arterial segments included the celiac trunk (50), hepatic artery (29), superior mesenteric artery (35), and other segments (4). In the PD group, SMA resection was the most common (31/51), the others were: CT (1/51), and hepatic artery or its branches (17/51). CT resection was the most frequent in the DP group (44/49), in one patient (1/49) CT + SMA resection was performed. No graft was used in cases of resection of a limited arterial segment (<3 cm) with the possibility of direct connection of the vessels extremities together. Otherwise, the autologous GSVs were preferentially harvested by separate groin incisions. In 105 patients, an associated venous resection (89%) was performed. According to the type of reconstruction, 61 patients had an arterial reconstruction with a graft interposition (56 saphenous autologous grafts, 4 autologous splenic arterial grafts, and 1 PTFE). Multiple arterial reconstructions were performed on 22 patients (18.6%). The authors compared results between two earlier (1990–2008) and later (2009–2018) periods. A comparative analysis of these two periods showed that the number of P-AR cases increased from 2.2% to 10.3% of the overall pancreatic resections performed (*p* = 0.0001). Comparison of patients undergoing P-AR in the second period showed a statistically significant higher use of preoperative chemotherapy (12% vs. 91%; *p* = 0.0001), higher rate of venous occlusion (12% vs. 46%; *p* = 0.001), higher use of transitory mesenterico-portal shunt (TMPS) to decrease the risk of bleeding (1 vs. 32; *p* = 0.001), and an increased number of SMA resections (3 vs. 31; *p* = 0.02). The greater technical complexity in the second period was not associated with increased morbidity (44% vs. 40.8%; *p* = 1.00), mortality (7.7% vs. 4.34%; *p* = 0.60), or relaparotomy rates (16% vs. 9%; *p* = 0.27). There were also no differences in the median overall survival in survival rates between these two periods (13.63 vs. 14.88 months; *p* = 0.72). The authors explained the higher rate of arterial resections in the second period that this was a consequence of their increasing institutional experience with extended pancreatectomies, vascular resections, the very low rate of pancreatic fistula (PF) achieved with pancreaticogastrostomy (PG) reconstruction, and introduction of a new neoadjuvant chemotherapy (FOLFIRINOX). In this study, the overall mortality and morbidity were 5.1% and 41.5%, respectively. There were 84 (75.4%) patients receiving neoadjuvant chemotherapy, 105 (89%) simultaneous venous resections, and 101 (85.5%) arterial reconstructions. The OS was 59%, 13%, and 11.8% at 1, 3, and 5 years, respectively. The median OS after resection was 13.70 months (CI 95%: 11–18.5 months). In a multivariate analysis, R0 resection (HR: 0.60; *p* = 0.01) and venous invasion (HR: 1.67; *p* = 0.04) were independent prognostic factors. This study revealed that in high-volume specialized centers, P-AR for locally advanced PDAC can be performed safely with limited mortality and morbidity [56].

In 2016, Perinel et al. [58] reported PDAC patients undergoing pancreatectomy between 2008 and 2014. The patients were divided into three groups as follows: group 1, pancreatectomy without vascular resection (*n* = 66); group 2, pancreatectomy with isolated venous resection (*n* = 31); and group 3, pancreatectomy with arterial resection (*n* = 14). In group 3 pancreatic resections were as follows: PD (*n* = 3), DP (*n* = 1), SP (*n* = 1), and TP (*n* = 9). The duration of operation was the longest in the group with arterial resection as follows: 4:50 ± 1:15 h, 5:30 ± 0:45 h, 6:20 ± 1:15 h in groups 1–3 respectively. Statistical difference was noted between groups 1 and 2 (*p* = 0.012). Intraoperative blood loss was comparable in all groups; however, intraoperative blood transfusion was the most frequent in group 3 (64% of patients) (G1 vs. G3 = 0.006, and G2 vs. G3 *p* = 0.009). The morbidity and mortality rates were comparable. The authors concluded that planned arterial resection in PDAC patients can be performed safely with a good outcome in highly selected patients. The crucial elements for defining the resectability was based on the extent of the axial arterial encasement with two criteria: the origin of the CT and SMA are free from tumor invasion and the possibility of distal reconstruction [58].

In 2017, Podda et al. [59] published the results of retrospective cohort analysis of patients undergoing PD between 2004 and 2014. All patients (*n* = 92) undergoing PD without vascular resection (PD, *n* = 72), with venous resection PD + VR (*n* = 16), with both arterial and venous resection (PD + VR + AR, *n* = 4) were included in the study. Authors reported similar morbidity rate and survival in PD with and without venous resection. Median survival for standard PD and PD with venous resection was 21 months and 18 months, respectively (*p* = 0.588). Patients undergoing PD with venous and arterial resection had a worse median survival of 7 months, compared with standard PD (*p* = 0.044). Median survival in the palliative bypass group was 4 months, comparable to PD with venous and arterial resection (*p* = 0.191). However, authors noted that there was no benefit of additional arterial resection in their experience [59].

In 2019, Loveday et al. [60] published the results of a study including 31 patients undergoing pancreatectomy and 46 patients undergoing explorative laparotomy for PDAC from 2009 to 2016. Patients undergoing resection (*n* = 31) were divided into two groups: with arterial resection (AR, *n* = 11) and without arterial resection (NAR, *n* = 20). There were 16 PD, 2 DP, and 2 TP in AR, and 11 PD without DP or TP in NAR groups. Arterial reconstruction was performed either with an end-to-end anastomosis or with an interposition graft. In patients following arterial resection, longer operative time (681 vs. 563 min, *p* = 0.006) and higher blood loss (1600 vs. 575 mL, *p* = 0.0004) was noted. Blood transfusion (*p* = 0.20), overall morbidity rate (*p* > 0.99), pancreatic fistula (*p* = 1.0), length of stay (*p* = 0.7), reoperation rate (*p* = 0.38), and mortality rate (*p* > 0.99) were similar in AR and NAR groups. Postoperative 90-day mortality was 3.2%. Median OS was comparable in both with and without arterial resection groups (19.7 for AR vs. 28.4 months for NAR, *p* = 0.41). Based on these findings the authors concluded that AR had comparable clinical and oncologic outcomes to NAR [60].

In 2018, Del Chiaro et al. [61] published the results of retrospective analysis of a cohort of operated borderline or locally advanced pancreatic cancer patients with surgically confirmed arterial involvement. Patients were divided into two groups: pancreatectomy with arterial resection (PAR) (±venous resection) (group 1, *n* = 34) and palliative surgery (Group 2, *n* = 39). Twenty-three patients (67.7%) in group 1 underwent combined artery-vein resection (AVR). For resection of the HA, reconstruction was performed by an end-to-end anastomosis or rotation of the SA. For reconstruction of the CT, an autologous or synthetic interposition graft was used, except when an Appleby procedure was performed. In the PAR group the following pancreatectomy types were performed: TP (*n* = 23) (68%), PD (*n* = 9) (26%), and DP (*n* = 2) (5.9%). The arterial resection types were the following: hepatic artery (HA)/celiac trunk (CT), 32 (55%); SMA, 3 (5.2%); and SMV/PV, 23 (40%). The types of vascular reconstruction were the following: For HA, CT (*n* = 32): end-to-end anastomosis, 15 (47%); anastomosis on GDA stump, 2 (6.3%); rotation of SA, 10 (31%); autologous interposition graft, 3 (9.5%); PTFE interposition graft, 1 (3.0%); ligature without reconstruction (Appleby procedure), 1 (3.0%); for SMA (*n* = 3): end-to-end anastomoses, 3 (100%); and for SMV/PV (*n* = 23): end-to-end anastomoses, 23 (100%). Operation time (426 ± 14 vs. 171 ± 11 min) and hospital stay (18 ± 2.4 vs. 9.3 ± 0.8 days) were longer and blood loss (613 ± 72 vs. 188 ± 21 mL) was higher in group 1 compared with group 2 (*p* < 0.0001). The postoperative mortality (2.9% vs. 2.6%, *p* = 0.9) and postoperative morbidity (38.2% vs. 25.6%, *p* = 0.2) were similar. The 1-, 3-, and 5-years survival in group 1 was superior to group 2 (63.7%, 23.4%, and Q3 23.4% vs. 41.7%, 3.2% and 0, *p* = 0.003). The authors concluded that PAR was safe and feasible with superior survival to palliation in well-selected patients [61].

In 2018, Tee et al. [62] published a study on pancreatectomies with AR (from 1990 to 2017). A total of 111 patients underwent pancreatectomy with AR including HA (54%), CT (44%), SMA (14%), or multiple ARs (14%), with revascularization in 55%. Arterial resections were classified into the following categories based on the extent of AR and complexity of revascularization: lateral AR and angioplasty only, en bloc segmental AR without formal revascularization, en bloc segmental AR with primary end-to-end anastomosis, and en bloc segmental resection with graft/conduit interposition revascularization. Interposition grafts/conduits used for arterial revascularization included both autologous (e.g., GSV, SA, and SFA) and synthetic material (e.g., Dacron, PTFE). Reconstructed arteries were routinely protected with pedicled round ligament or omental patches from surrounding viscera, if available, to prevent contamination. PD comprised 41% of the pancreatectomies, with 41% being subtotal/extended DP and the remaining 18% TP. There were 60 resections of any HA (common, proper, right or left, replaced HA, or combination), 49 CA resections, 15 SMA resections, and 15 patients who underwent en bloc multiple AR. The four types of AR were identified: 11% underwent lateral resection and angioplasty, 34% underwent AR without formal revascularization, 30% underwent AR with primary end-to-end anastomosis, and 25% underwent AR with graft/conduit interposition revascularization. The authors concluded that pancreatectomy with AR was associated with a higher risk compared with standard pancreatectomy. The authors noted that mortality has decreased in the modern era; however, morbidity remained high from hemorrhagic, fistula, or ischemia-related complications [62].

In 2020, Kwon et al. [63] published a study conducted on 109 PDAC patients undergoing pancreatectomy with AR between 2000 and 2017. Among 109 patients, 38 underwent surgery after neoadjuvant chemotherapy, and 71 underwent upfront surgery. Operation types were as follows: PD 42 (38.5%), DP 47 (43.1%), and TP 20 (18.3%). Arial resections included for PD/DP/TP: HA 54 (31/6/17), SMA 10 (10/0/0,) and CT 45 (1/41/3). Vascular reconstructions were the following for HA/SMA/CT: end-to-end anastomosis, 46 (37/7/2); wedge resection with primary repair or patch, 5 (4/1/0); interposition graft, 13 (9/2/2); and non-anastomosis, 45 (4/0/41). Concurrent vein resection (PV or SMV) was performed in 62 (56.9%) patients. The major morbidity (≥grade III) and mortality rates were 26.6% and 0.9%, respectively. The median OS of all patients was 18.4 months. The authors concluded that pancreatectomy with AR can be an acceptable procedure in terms of postoperative morbidity and mortality in highly selected patients. Additionally, the survival benefit was noted in patients who were undergoing neoadjuvant chemotherapy compared with patients undergoing upfront surgery [63].

In 2017, Beane et al. [64] analyzed the outcomes of PD with and without vascular resection in a large, multicenter cohort. This study included patients from 43 institutions as part of the American College of Surgeons National Surgical Quality Improvement Program (ACS-NSQIP) Pancreatectomy Demonstration Project. Over a 14-month period, 1414 patients underwent PD without (82.2%) or with major venous (PD + VR; 13.7%) or arterial (PD + AR; 4.0%) vascular resection. Postoperative morbidity and mortality following PD + AR (51.0% and 3.6%) was comparable to PD + VR (46.9% and 3.6%) and PD (44.3 and 1.5%, *p* = 0.50 and 0.43). The analysis revealed that vascular resection was associated with a significantly longer duration of operation (7:37 vs. 6:11, *p* < 0.01), blood transfusion (42.2% vs. 18.1%, *p* < 0.01), deep venous thromboembolism (6.9% vs. 0.9%, *p* < 0.01), postoperative septic shock (6.9% vs. 1.7%, *p* = 0.02), and length of stay (12.2 vs. 10 days, *p* = 0.01); while overall morbidity (45.7% vs. 46.6%, *p* = 0.53) and mortality (1.0% vs. 0%, *p* = 0.07) were comparable. The authors concluded that PD with vascular resection was associated with a longer duration of operation, higher rate of perioperative transfusions, deep venous thrombosis, septic shock, as well as longer duration of hospitalization, but similar morbidity and mortality compared with PD without vascular resection [64].

A meta-analysis by Małczak et al. published in 2020 revealed a higher risk of mortality (Risk Ratio, RR: 4.09; *p* < 0.001) and postoperative morbidity (RR: 1.4; *p* = 0.01) and comparable rate of pancreatic fistula, biliary fistula, cardiopulmonary complications, duration of hospitalization and non-R0 rate. In addition, a worse 3-year survival was noted in patients following pancreatectomy with arterial resection.

#### Conclusions

The rate of arterial resections during pancreatectomy in PDCA patients was increased. Currently, there is no significant benefit of arterial resection during pancreatectomy for PDCA. Taking into account a higher rate of morbidity and mortality and similar survival compared with standard pancreatectomy, arterial resection should be performed only in highly selected patients.

## 5. Difficulties in Selection of Patients for Pancreatectomy with Concurrent Vascular Resection in The Era of Neoadjuvant Therapy

The 2020 NCCN guidelines recommend neoadjuvant therapy (NAT) including FOLFIRINOX or gemcitabine plus nab-paclitaxel (G-nP) for borderline resectable/locally advanced pancreatic ductal adenocarcinoma (BR/LA PDAC) to downsize tumors to the point where they may be resected [65]. FOLFIRINOX significantly improved the prognosis of patients with LA PDAC, but long-term survival is noted only after pancreatectomy [66]. The introduction of NAT caused an increase in pancreatectomy with concomitant vascular resections in the last decade. Therefore, correct interpretation of these results is very important and difficult. Currently, it is well-known that invasion of the porto-mesenteric venous axis by PDCA is not considered a contraindication for resection, but expert consensus defined the extension to the hepatic, coeliac, and superior mesenteric arteries as indicating a locally advanced “unresectable” tumor [67].

Although venous resections during pancreatectomy are currently a standard treatment, the feasibility of arterial resection is heavily discussed, yet vascular resections in general are not always necessary in resected patients with LAPC following NAT (18–80.8%). Arterial resections represent up to 35% of the resected cases and about 60% of these had histologic arterial infiltration. That is because it is not possible to differentiate viable tumor and post-treatment fibrosis after FOLFIRINOX using the radiological investigations after FOLFIRINOX. A study by Ferrone et al. [68] showed that despite of unresectability according to radiological imaging, tumor removal without tumoral involvement of the resection margin and with a significantly improved prognosis was possible in 92% of patients [68]. Radiological tumor regression, according to the RECIST criteria, after NAT for PDCA is noted in about 11%, while the stable disease in reported in 83% of them. Therefore, radiological re-assessment after NAT is not useful to describe the possibility of resection. Frequently, following NAT, the separation of the arteries from the surrounding tissue, without the necessity for resection, is observed [67].

Current NAT is not related to the problem of local recurrence, controlling the distant recurrence but is the problem, because it is significantly more frequently noted. According to the literature, local recurrence was noted in only 20–24% of patients, and recurrence at distant sites was reported in 73–80% of patients with BRPC/LAPC after resection. Therefore, the selection of candidates for resection following NAT having a lower risk of distant recurrence, is more difficult. Currently, FOLFIRINOX is the chemotherapy that can lead the most patients to resection and shows a predicted 5-year survival of 40%. It was observed that surgery is associated with survival benefit even after using other, more tolerable, combination regimens or after dose reductions in the potent FOLFIRINOX that increases a number of patients who may benefit from multimodality therapy. Factors associated with improved survival were explored after NAT, and most often they can be found on the final histopathological finding as the degree of tumor regression in response to the cytotoxic drugs. This information is available only after surgery; therefore, it cannot be used as a selection tool before surgery. CA19-9 is the only preoperative factor that so far was related to survival. Currently, there is no consensus regarding a level which should be used as the cut-off value when resection should be recommended. Complete normalization or levels <100 were suggested. Pancreatectomy may be associated with survival benefit for all values of elevated CA19-9 compared with no resection. Balance between the magnitude of the survival gain and the negative consequences of extended resection is required during decision-making [66]. Generally, the expected benefits should outweigh the potential risks associated with a complex operation.

## 6. Summary and Conclusions

All above-mentioned articles regarding pancreatectomy with concurrent venous and arterial resections in PDCA patients are summarized in Table 1. PD with vascular resection is increasingly used in the treatment of borderline and locally advanced PDAC. Venous resection currently is a standard treatment and it is recommended for R0 resection due to comparable morbidity and mortality with standard PD. Whereas arterial resection and reconstruction still remains controversial due to significantly increased rates of postoperative morbidity. Based on the literature review, arterial resection and reconstruction may be appropriate in selected patients. Novel systemic neoadjuvant treatment regimens, such as FOLFIRINOX, are also very important in the treatment of advanced PDAC. Therefore, multidisciplinary management should be used in order achieve R0 resection and improve survival in patients. In addition, consensus regarding perioperative anticoagulation in patients undergoing pancreatectomy with concomitant vascular resections is needed.

## Figures and Tables

**Table 1 cancers-14-01193-t001:** Summary of studies on vascular resections in pancreatectomy.

Reference	Patients No.	Study Design	Study Results and Conclusions
Venous Resections			
Shao et al. [12]	146	Retrospective study, single-center comparison of FL (13) venoplasty with other reconstructions	The shortest hospital stay, operation duration, lowest blood loss in FL group The highest (100%) patency rate in FL group The lowest antiplatelet/anticoagulation proportion in FL group Comparable morbidity and log-term survival rates in all groups
Shao et al. [19]	6	Retrospective study, single-center FL use as lateral patch for MPV	Autologous FL graft is a safe lateral substitute for MPV reconstruction 100% R0 resection, 100% patency rate
Zhiving et al. [20]	10	Retrospective study, single-center a patch graft (*n* = 6) and a conduit graft (*n* = 4)	FL grafts might be considered for reconstruction of PV/SMV in the absence of appropriate vascular grafts
Malinka et al. [21]	11	Retrospective study, single-center FL use as reconstructions following wedge (*n* = 9) and segmental (*n* = 2) venous resections	Mortality rate 0% Patency rate 81.81% FL graft is a safe tissue for venous reconstruction
Dokmak et al. [22]	30	Retrospective study, single-center pancreatic (*n* = 18) or hepatic (*n* = 12) resections with venous reconstruction using PP	III Clavien grade in four (13%) patients R0 resection 100% Patency rate 97% PP is a safe lateral patch for venous reconstructions in hepatobiliary resections
Dokmak [23]	52	Prospective study, single-center pancreatic (*n* = 18) or hepatic (*n* = 12) resections with venous reconstruction using PP	>III Clavien grade in eight (15%) patients Patency rate 96, 100% (for lateral patches), 33% for tubular graft PP is a rapid, inexpensive, and safe vascular graft, therapeutic coagulation is unnecessary
Lee et al. [24]	34	Retrospective study, single-center PV/SMV reconstructions using GSV or FV	GSV and FV can be used for venous reconstruction in PD with minimal complications rate and late mortality, and high patency Long-term survival comparable with PD without venous reconstruction
Turley et al. [25]	204	Retrospective study, single-center comparison of PD with VR (*n* = 42) and without VR (*n* = 162)	Higher median blood loss in PD with VR Similar mortality and morbidity rates, duration of hospitalization, and readmission rates in both groups Graft patency was 91.7%
Krepline et al. [26]	43	Retrospective study, single-center pancreatic resections with venous reconstructions	Graft patency was 91% Optimal prevention of vascular thrombosis is needed
Hirono et al. [27]	128	Retrospective study, single-center TP (*n* = 5), PD (*n* = 99), and DP (*n* = 24) with PV/SMV resection including grafts (*n* = 14)	IJV is superior to EIV in venous reconstruction (no complications)
Glebova et al. [28]	173	Prospective study, single-center	Long duration of operation and use of prosthetic grafts for venous reconstruction are risk factors for postoperative PV thrombosis
Dua et al. [29]	90	Retrospective study, single-center PV/SMV resection/reconstruction during a pancreatectomy using different techniques	EE and TV should be preferred reconstructions due to the highest patency rate
Terasaki et al. [30]	199	Retrospective study, single-center 199 PD including 122 PD with PVR (EE and interposition graft using right EIV	Longer survival in patients following PD without VR Similar survival in patients following EE and interposition graft Interposition graft using the right EIV for PVR following PD was safe and effective
Pantoya et al. [15]	18	Retrospective study, single-center 18 PD with venous interposition graft reconstructions: GSV (*n* = 13), and IJV (*n* = 5)	GSV and IJV are comparable for venous reconstruction in PD
Chan et al. [31]	76	Retrospective study, single-center PD and TP with venous reconstructions	1-year primary patency of primary repair is superior to EE and interposition graft, and it should be preferred if it possible
Labori et al. [32]	603	Systematic review comparison of four graft types: autologous vein, autologous parietal peritoneum/falciform ligament, allogeneic cadaveric vein/artery, synthetic grafts	The early and overall graft thrombosis rate: 7.5% and 22.2% for synthetic graft, 5.6% and 11.7% for autologous vein graft, 6.7% and 8.9% for autologous parietal peritoneum/falciform ligament, and 2.5% and 6.2% for allograft
Pan et al. [33]	118	Retrospective study, single-center 58 PD with SMV/PVR	Significantly better survival in patients with shorter resections, but comparable short-term postoperative results regardless venous resection length
Fui et al. [36]	810	Retrospective study, single-center 147 PD with SMV/PVR	Length of SMV/PV resection ≥ 31 mm as independent predictor of medium-term, severe anastomotic stenosis
Kim et al. [37]	249	Retrospective study, single-center 66 PV-PD, including 27 (41%) planned and 39 (59%) unplanned PV resections.	Planned PV resections associated with higher rates of postoperative major and vascular complications and higher R0 resection rates compared with unplanned resections compared with unplanned resections Comparable survival in both groups
Cheung et al. [42]	78	Retrospective study, single-center 46 standard PD and 32 PD with SMV/PVR	Comparable perioperative morbidity, mortality rate and survival in both groups
Selvaggi et al. [43]	60	40 PD with SMV/PVR and 20 palliative by-passes	Longer survival in resection group compared with by-passes
Yu et al. [11]	2890	Meta-analysis comparison of standard PD and PD with SMV/PVR	Comparable perioperative morbidity, mortality, and 1-year, 3-year survival in two groups
Murakami et al. [44]	937	Retrospective study, multi-center 502 standard PD and 435 PD with SMV/PVR	Comparable perioperative morbidity, mortality in two groups PV/SMV resection not independent prognostic factor for OS
Jeong et al. [45]	276	Retrospective study, single-center 230 standard PD and 46 PD with SMV/PVR	Comparable short-term and long-term results in both groups
Wang et al. [46]	208	Retrospective study, single-center 166 standard PD and 42 PD with SMV/PVR	Comparable survival time and R0 resection in both groups
Delpero et al. [47]	1399	Retrospective study, multi-center 997 standard PD and 402 PD with SMV/PVR	Comparable postoperative morbidity and mortality in both groups Longer survival in standard PD Venous resection as independent poor prognostic factor for survival
Serenari et al. [48]	99	Retrospective study, single-center PD + TVR (25.3%), 12 to PD + SVR (12.1%), 23 to TP + TVR (23.2%), and 39 to TP + SVR (39.4%).	Comparable short-term results Higher median OS in patients undergoing PD + TVR compared with TP+SVR (29.5 vs. 7.9 months) TP and SVR are independent poor prognostic factors for OS
Giovinazzo et al. [49]	9005	Meta-analysis comparison of standard pancreatectomy with SMV/PVR	Increased postoperative mortality, higher rates of non-radical surgery and worse survival after pancreatic resection with PV-SMV resection
Peng et al. [50]	12031	Meta-analysis 9845 standard PD and 2186 PDVR	Higher risk of morbidity and mortality, longer duration of hospitalization, and a lower R0 resection rate in PDVR
Fancellu et al. [51]	6037	Meta-analysis 71.4% standard PD and 28.6% PDVR	Comparable morbidity, higher 30-day mortality, and lower 1-, 3-, 5-year OS in PDVR
Filho et al. [52]	2986	Meta-analysis comparison of standard PD and PDVR	PDVR associated with a higher risk for postoperative morbidity and mortality, longer duration of hospitalization and higher blood loss as well as worse OS compared with standard PD
Arterial Resections			
Ouaissi et al. [54]	149	Retrospective study, single-center 82 standard PD/TP, 67 PD/TP with isolated venous resection, 8 PD/TP with arterial (SMA, CHA, right hepatic artery (RHA)) and/or venous resection	Higher duration of operation and blood loss in vascular resection groups Comparable postoperative morbidity and mortality and duration of hospitalization in all groups Higher R1 resection rate and worse 10-year OS and DFS in vascular resection groups
Bockhorn et al. [55]	518	Retrospective study, single-center 29 AEBR, 449 standard pancreatectomies, 40 bypasses	Higher morbidity and mortality rate in AEBR compared with standard pancreatectomy Additional portal vein resection is an independent predictor of morbidity The highest duration of operation and hospital stay in AEBR OS comparable in AEBR and standard resection, OS lower in bypass group
Bachelier et al. [56]	52	Retrospective study, single-center 26 AR+ and 26 AR − PD/DP	Comparable short-term results (morbidity, mortality), duration of operation, and 1-and 3-year survival are comparable in both groups
Bachelier et al. [57]	118	Retrospective study, single-center comparison of PD/DP/TP with AR in earlier (1990–2008) and later (2009–2018) periods	Higher rate of AR, higher use of neoadjuvant chemotherapy. Venous occlusion, transitory mesenterico-portal shunt, increased number of SMA resections in later periods (increase from 2.2% to 10.3%) Comparable morbidity and mortality rate and OS in two periods
Perinel and et al. [58]	111	Retrospective study, single-center 66 pancreatectomies without vascular resection, 31 pancreatectomies with isolated venous resection, 14 pancreatectomies with arterial resection	The longest duration of operation in AR group Comparable blood loss, morbidity and mortality in all groups
Podda et al. [59]	92	Retrospective study, single-center 72 PD without vascular resection, 16 PD with isolated venous resection, 4 pancreatectomies with venous and arterial resection	Worse survival in PDAR compared with standard PD Comparable survival in PDAR and bypass group No benefit of additional arterial resection
Loveday et al. [60]	31	Retrospective study, single-center 11 standard pancreatectomies, 20 pancreatectomies with AR	Longer operative time and higher blood loss in AR Comparable blood transfusion, overall morbidity rate, pancreatic fistula, length of stay, reoperation rate, and mortality rate, and OS in both groups
Del Chiaro et al. [61]	61	Retrospective study, single-center 39 palliative surgery, 34 pancreatectomies with AR	Comparable morbidity and mortality rate in both groups. Longer hospital stay, duration of hospitalization and higher blood loss in AR compared with palliative group Longer 1-, 3-, 5-year survival in SR compared with palliative group
Tee et al. [62]	111	Retrospective study, single-center 111 pancreatectomies with various AR	Higher risk of complications in pancreatectomy with AR compared with standard pancreatectomy
Kwon et al. [63]	109	38 pancreatectomies after neoadjuvant chemotherapy, and 71 upfront surgeries	Major morbidity (≥grade III) 26.6%, mortality 0.9%. OS was 18.4 months Better survival after neoadjuvant chemotherapy than in upfront surgery
Beane et al. [64]	1414	Retrospective study, multi-center Standard PD (82.2%), PD with major venous resection (13.7%), PD with major arterial resection	Comparable postoperative morbidity and mortality in all groups Longer duration of operation, longer duration of hospitalization, higher rate of blood transfusion, deep venous thromboembolism, postoperative septic shock in vascular resection groups
Małczak et al. [53]	2710	Meta-analysis of 19 studies on arterial resection in pancreatectomy	Higher mortality and morbidity in pancreatectomy with AR Comparable rate of pancreatic fistula, biliary fistula rate, cardiopulmonary complications, length of hospital stay and non-R0 rate Worse 3-year survival in pancreatectomy with AR

PD: pancreaticoduodenectomy; DP: distal pancreatectomy; TP: total pancreatectomy; FL: falciform ligament; MPV: mesentericoportal vein; PP: parietal peritoneum; GSV: great saphenous vein; FV: femoral vein; IJV: internal jugular vein; EIV: external iliac vein; EE: end-to-end anastomosis; TV: transverse venorrhaphy; TVR: tangential venous resection; SVR: segmental venous resection; OS: overall survival; DFS: disease-free survival; AEBR: arterial en bloc resection; VR: venous resection; AR: arterial resection.

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
