# Peer review of "Vascular Resection in Pancreatectomy—Is It Safe and Useful for Patients with Advanced Pancreatic Cancer?"

_cancers, 2022, doi:10.3390/cancers14051193_

Round 1

Reviewer 1 Report

Congratulations to the authors for such an interesting article. As the authors mention, surgery remains the only possibility of treatment with curative intent in patients with pancreatic cancer, and a safe vascular resection (mainly in cases of venous tumor involvement) offers an opportunity to some of our patients.
The review of the subject is current and profound, with an interesting analysis of vascular permeability after reconstruction, according to the technique used.
My only review comment concerns the bibliography. Please, I ask the authors to review citations 25, 29, 48 and 49, where there are minor typographical errors.

Author Response

Dear Editors,

Dear Reviewers,

 Thank you for peer reviewing of our manuscript cancers-1607465, entitled " Vascular Resection in Pancreatectomy - Is It Safe and Useful for Patients with Advanced Pancreatic Cancer?".

Thank you for your questions and comments. We have fully addressed all the comments and my responses appear below. Our revised work includes corrections according to reviewers’ comments in the text. The changes, made according to reviewers’ comments, are highlighted in red print in the text.

We take this opportunity to express my gratitude to the reviewers for their constructive and useful remarks. Their comments allowed us to identify areas in my manuscript that needed modification.

We also thank you for allowing me to resubmit a revised copy of the manuscript.

We hope that the revised manuscript is now acceptable for publication in Cancers.

Yours sincerely,

Beata Jabłońska.

Responses to Reviewers comments

Reviewer 1

Comment:

Congratulations to the authors for such an interesting article. As the authors mention, surgery remains the only possibility of treatment with curative intent in patients with pancreatic cancer, and a safe vascular resection (mainly in cases of venous tumor involvement) offers an opportunity to some of our patients.
The review of the subject is current and profound, with an interesting analysis of vascular permeability after reconstruction, according to the technique used.
My only review comment concerns the bibliography. Please, I ask the authors to review citations 25, 29, 48 and 49, where there are minor typographical errors.

Answer:

Thank you for appreciating our work. Thank you very much for your positive opinion.

Citations 25, 29, 48 and 49 have been reviewed and minor typographical errors have been corrected.

Reviewer 2 Report

This is a comprehensive review of vascular resection during pancreatectomy for pancreatic cancer. The topic is interesting, the literature search is extensive and the results of the study confirm previous that venous resection is justified for R0 resection and arterial resection is not recommended although some Authors continue to perform such operation. Vascular resection represents a complex operation increasingly performed in the last decade probably after the introduction of new multidrugs therapy (FOLFIRINOX, abraxane, ecc.). Correct interpretation of these results is difficult. I think the Authors should discuss this point together with the explanation of the term "selected patients" in which arterial resection could be useful.

Author Response

Dear Editors,

Dear Reviewers,

 Thank you for peer reviewing of our manuscript cancers-1607465, entitled " Vascular Resection in Pancreatectomy - Is It Safe and Useful for Patients with Advanced Pancreatic Cancer?".

Thank you for your questions and comments. We have fully addressed all the comments and my responses appear below. Our revised work includes corrections according to reviewers’ comments in the text. The changes, made according to reviewers’ comments, are highlighted in red print in the text.

We take this opportunity to express my gratitude to the reviewers for their constructive and useful remarks. Their comments allowed us to identify areas in my manuscript that needed modification.

We also thank you for allowing me to resubmit a revised copy of the manuscript.

We hope that the revised manuscript is now acceptable for publication in Cancers.

Yours sincerely,

Beata Jabłońska.

Responses to Reviewers comments

Reviewer 2

Comment:

This is a comprehensive review of vascular resection during pancreatectomy for pancreatic cancer. The topic is interesting, the literature search is extensive and the results of the study confirm previous that venous resection is justified for R0 resection and arterial resection is not recommended although some Authors continue to perform such operation. Vascular resection represents a complex operation increasingly performed in the last decade probably after the introduction of new multidrugs therapy (FOLFIRINOX, abraxane, ecc.). Correct interpretation of these results is difficult. I think the Authors should discuss this point together with the explanation of the term "selected patients" in which arterial resection could be useful.

Answer:

Thank you for appreciating our work. Thank you very much for your positive opinion.

The paragraph, presenting discussion on correct interpretation of the results of neoadjuvant therapy (NAT) together with explanation of the term "selected patients" in which arterial resection, has been added as follows:

  1. Difficulties in selection of patients for pancreatectomy with concurrent vascular resection in the era of neoadjuvant therapy

The 2020 NCCN guidelines recommend neoadjuvant therapy (NAT) including FOLFIRINOX or gemcitabine plus nab-paclitaxel (G-nP) for borderline resectable/locally advanced pancreatic ductal adenocarcinoma (BR/LA PDAC) to downsize tumors to the point where they may be resected [65]. FOLFIRINOX significantly improved the prognosis of patients with LA PDAC, but long-term survival is noted only after pancreatectomy [66]. The introduction of NAT has caused an increase of pancreatectomy with concomitant vascular resections in the last decade. Therefore, correct interpretation of these results is very important and difficult. Currently, it is well known that invasion of the porto-mesenteric venous axis by PDCA is not considered a contraindication for resection, but expert consensus has defined the extension to the hepatic, coeliac, and superior mesenteric arteries as indicating a locally advanced “unresectable” tumor [67].

Although venous resections during pancreatectomy are currently a standard treatment, the feasibility of arterial resection is heavily discussed, yet vascular resections in general are not always necessary in resected patients with LAPC following NAT (18–80.8%). Arterial resections represent up to 35% of the resected cases and about 60% of these had histologic arterial infiltration. That is because it is not possible to differentiate viable tumor and post-treatment fibrosis after FOLFIRINOX using the radiological investigations after FOLFIRINOX. A study by Ferrone et al. [68] showed that despite of unresectability according to radiological imaging, tumor removal without tumoral involvement of the resection margin and with significantly improved prognosis was possible in 92% of patients [68]. Radiological tumor regression, according to the RECIST criteria, after NAT for PDCA is noted in about 11%, while the stable disease in reported in 83% of them. Therefore, radiological re-assessment after NAT is not useful to describe the possibility of resection. Frequently, following NAT, separation of the arteries from the surrounding tissue, without the necessity for resection, is observed [67].

Current NAT is not related to the problem of local recurrence, controlling the distant recurrence but is the problem, because it is significantly more frequently noted. According to the literature, local recurrence was noted in only 20–24% of patients, and recurrence at distant sites was reported in 73–80% of patients with BRPC/LAPC after resection. Therefore, selection of candidates for resection following NAT having a lower risk of distant recurrence, is the more difficult. Currently, FOLFIRINOX is the chemotherapy that can lead the most patients to resection and shows predicted 5-year survival of 40%. It has been observed that surgery is associated with survival benefit even after using other, more tolerable, combination regimens or after dose reductions of the potent FOLFIRINOX that increases a number of patients who may benefit from multimodality therapy. Factors associated with improved survival have been explored after NAT, and most often they can be found on the final histopathological finding as the degree of tumor regression in response to the cytotoxic drugs. This information is available only after surgery. Therefore, cannot be used as a selection tool before surgery. CA19-9 is the only preoperative factor that so far has been related to survival. Currently, there is no consensus regarding a level which  should be used as the cut-off value when resection should be recommended. Complete normalization or levels <100 have been suggested. Pancreatectomy may be associated with survival benefit for all values of elevated CA19-9 compared to no resection. But the balance between the magnitude of the survival gain and the negative consequences of extended resection is required during making decision [66]. Generally, the expected benefits should outweigh the potential risks associated with a complex operation.
